# Uncertainty about social interactions leads to the evolution of social heuristics

Pieter van den Berg [1] & Tom Wenseleers [1]

Individuals face many types of social interactions throughout their lives, but they often cannot perfectly assess what the consequences of their actions will be. Although it is known that unpredictable environments can profoundly affect the evolutionary process, it remains unclear how uncertainty about the nature of social interactions shapes the evolution of social behaviour. Here, we present an evolutionary simulation model, showing that even intermediate uncertainty leads to the evolution of simple cooperation strategies that disregard information about the social interaction ('social heuristics'). Moreover, our results show that the evolution of social heuristics can greatly affect cooperation levels, nearly doubling cooperation rates in our simulations. These results provide new insight into why social behaviour, including cooperation in humans, is often observed to be seemingly suboptimal. More generally, our results show that social behaviour that seems maladaptive when considered in isolation may actually be well-adapted to a heterogeneous and uncertain world.

---

[1] KU Leuven Lab of Socioecology and Social Evolution, Naamsestraat 59, 3000 Leuven, Belgium. Correspondence and requests for materials should be addressed to P.v.d.B. (email: pvdberg1@gmail.com)

Cooperation, where individuals incur a personal cost to provide a benefit to others, is a common phenomenon in the natural world, including in humans. Over the decades, evolutionary models have been vital to our understanding of why cooperative behaviour can evolve despite its costs. For ease of analysis and interpretation, these models are typically formulated at a relatively abstract level and often consider the evolution of behaviour in a single isolated cooperation situation, such as the (iterated) prisoner's dilemma game[1–4]. Although this has been an instructive approach, it neglects the fact that individuals typically face many different types of social situations throughout their lives, which they cannot always perfectly distinguish from each other.

In a world that is heterogeneous and uncertain, we might not expect that evolution perfectly tailors behavioural rules to each specific social context in isolation[5–10]. Rather, we might expect that evolution produces heuristics: simple behavioural rules that perform well across various different situations[11–13]. Previous work has shown that the existence of multiple interaction contexts can indeed lead to relatively simple strategies that are not perfectly fit for each interaction context in isolation, if there are explicit constraints on the flexibility of the cognitive machinery[14,15]. We also know that various sources of uncertainty can have a profound effect on the evolution of cooperation. For example, theory shows that both relatively small amounts of uncertainty about the behaviour of the interaction partner[16] and uncertainty about whether or not individuals will interact with the same interaction partner again in the future[8,9] can positively impact the evolution of cooperation. Also, empirically, it has been well documented that humans employ heuristic decision making[17–19], including in social situations[20–22]. However, it is not yet clear if uncertainty about the payoff consequences of social interactions can lead to the evolution of social heuristics, and, if so, which type of heuristics we should expect to evolve.

In this paper, we present a simulation model of the evolution of cooperative behaviour in a world in which many types of social interactions may arise, and in which individuals are uncertain about what kind of situation they are facing at any given time. In our simulations, the interaction types that individuals may face all have an element of cooperation: individuals must repeatedly decide whether or not to provide a benefit to their interaction partner. However, the exact payoff consequences of cooperating vary between interactions, ranging from situations where cooperation is always the best decision (regardless of the behaviour of the interaction partner), via various versions of the prisoner's dilemma game, to situations where it never pays to cooperate (Fig. 1a). All individuals have a behavioural strategy for the social interactions they encounter, determined by their genotype. This strategy can either be context-dependent, where the individuals take the available information about the nature of social interaction into account, or heuristic, where the individual disregards this information (Fig. 1c). Between different simulations, we systematically vary the degree of uncertainty that individuals have about the consequences of cooperation (Fig. 1b) and investigate how uncertainty affects (1) the evolution of heuristic decision making and (2) the level of cooperation in the population.

Our results show that under low uncertainty, evolution leads to context-dependent strategies: individuals use distinct substrategies depending on which social interaction type they believe they are facing. However, with increasing uncertainty, it becomes increasingly likely for evolution to produce heuristic strategies, which completely disregard the available information about the nature of the social interaction. Even under intermediate uncertainty, where individuals still have a fair amount of reliability in assessing the social interaction they find themselves in, social

heuristics are a highly likely outcome of evolution. In addition, we observe that the cooperation rate in populations that evolve under intermediate-to-high uncertainty is almost double the cooperation rate in low-uncertainty populations, in which more sophisticated strategies evolve. Indeed, the social heuristics that evolve even cooperate in situations where cooperation would never be expected if considered in isolation. These results indicate that seemingly maladaptive social behaviour may in fact be well-adapted to a world in which individuals face a spectrum of social interactions and are uncertain about the specifics of the interaction they are facing at any given time.

## Results

**The model**. We developed an individual-based evolutionary simulation model consisting of 1000 individuals per generation, who each have ten repeated social interactions with different interaction partners in their lifetime. Each of these interactions lasts for ten interaction rounds, in which both interaction partners simultaneously decide whether to cooperate or not. Cooperating results in a fixed benefit $b$ for the interaction partner ($b = 2$ in all simulations) but can have varying consequences $c$ for the acting individual. At the start of each repeated interaction, these consequences are randomly drawn from a uniform distribution ranging from a severe cost ($c = -3$) to a direct benefit ($c = 1$) (Fig. 1a). If $c < -2$, the cost of cooperating exceeds the benefit to the interaction partner. This means that defection does not only pay off better than cooperation regardless of the action of the interaction partner (i.e., defection is dominant), but mutual defection also pays off better than mutual cooperation. If $-2 < c < 0$, defection is still dominant, but mutual cooperation pays off better than mutual defection. Interactions in this range can be classified as variations of the prisoner's dilemma game. If $c > 0$, cooperation is dominant, and mutual cooperation ensures the best possible payoff.

In our simulations, individuals cannot directly observe what are the consequences of cooperating ($c$) in the interaction they are facing. Instead, they have a subjective perception of what the consequences of cooperating are (this perception is denoted as $c_p$). This perceived value is drawn from a beta distribution with mode $c$ (Fig. 1b). Between simulations, we systematically vary the degree of uncertainty by varying the variance of this beta distribution. If uncertainty = 0, the distribution is so narrow that individuals know exactly what kind of interaction they are facing ($c_p = c$); if uncertainty = 1, the distribution is equal to a uniform distribution over the entire range (hence, individuals have no reliable information at all about the consequence of cooperating). Between these extremes, we ran simulations for various degrees of intermediate uncertainty (in steps of 0.1), where uncertainty is directly proportional to the variance of the beta distribution from which $c_p$ is drawn.

In our model, all individuals have a genotype determining the strategy that they use in social interactions (see Fig. 1c for a schematic overview). Individuals can either implement a context-dependent strategy, which allows the individual to implement different substrategies (C1 or C2) depending on the perceived consequences of cooperation ($c_p$) of the current interaction, or a heuristic strategy, which is insensitive to the individual's perception of the interaction type at hand (always implementing substrategy H). Both the context-dependent and the heuristic strategies are coded in the individual's genotype, and a single Boolean locus ($S$) determines which of the strategies is used (acting as a switch). Another continuous locus ($T$) determines which of the context-dependent substrategies the individual implements, depending on the perceived consequence of cooperation of the present interaction. Specifically, if $c_p < T$, the

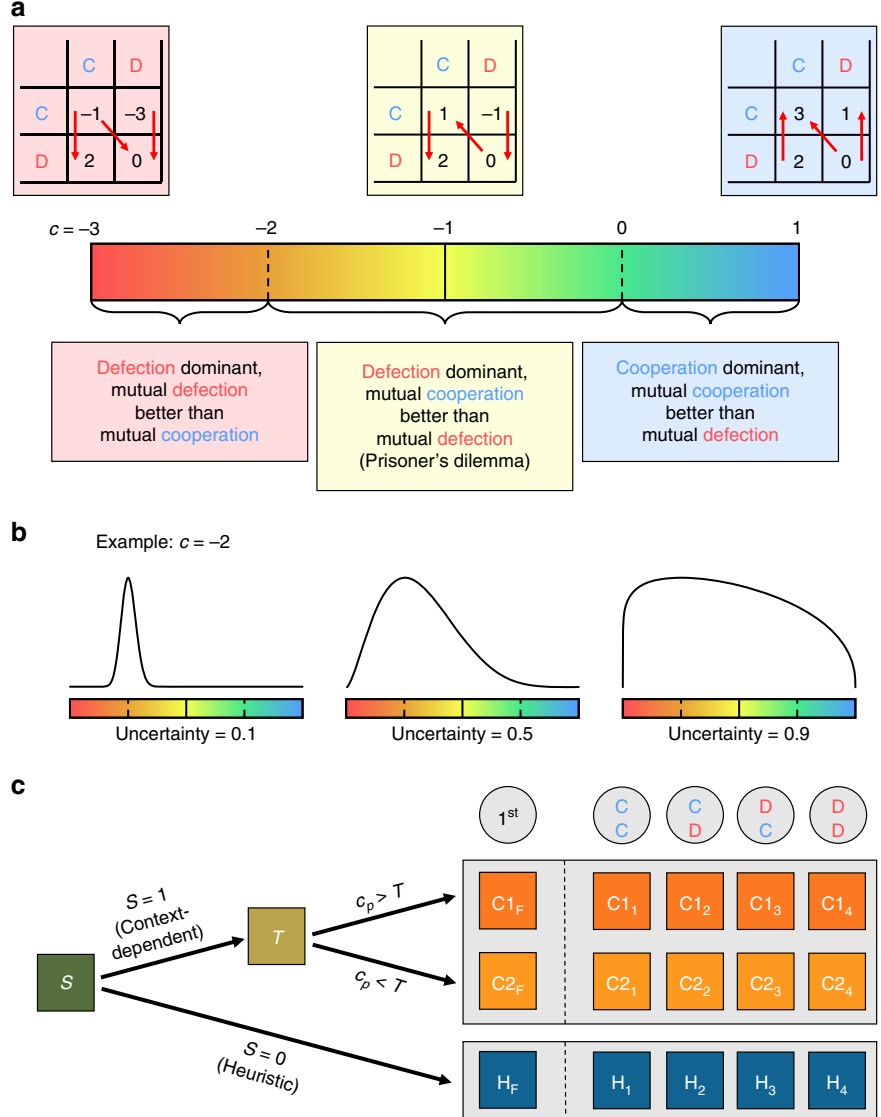

**Fig. 1** Implementation of **a** heterogeneity in social interactions, **b** uncertainty about social interactions and **c** cooperation strategies in our simulation model.
**a** Individuals face various social situations, always containing an element of cooperation. If an individual cooperates, she provides a fixed benefit to her interaction partner ($b = 2$), at a variable consequence ($c$) to herself. $c$ can take any value between $-3$ (a severe cost) and 1 (a benefit; horizontal bar). Payoff matrices show payoffs to the row player for the specific cases where $c$ is equal to $-3$ (red), $-1$ (yellow) and 1 (blue). **b** Uncertainty ($u$) determines how individuals perceive the value of $c$. Specifically, this perceived value $c_p$ is drawn from a beta distribution with mode $c$, and variance that is proportional to uncertainty. For $u = 0$, the variance of the distribution is 0 ($c_p = c$); for $u = 1$, the distribution is uniform. **c** Each individual carries 17 genes (indicated by squares). Gene $S$ determines whether the individual implements a heuristic ($S = 0$) or a context-dependent ($S = 1$) strategy. If $S = 0$, the individual always implements substrategy H (regardless of $c$). If $S = 1$, the individual implements either of her two context-dependent substrategies (C1 or C2), depending on the value of threshold gene $T$ and her perception of $c$ for the current situation ($c_p$). If $c_p > T$, the individual implements substrategy C1; if $c_p < T$, she implements substrategy C2. Each substrategy consists of five genes. The genes with subscript F determine the probability that the individual cooperates in the first round of the repeated interaction (these genes can take any continuous value between 0 and 1). The other four genes (subscripted 1–4) determine whether the individual defects (0) or cooperates (1), depending on the outcome of the previous interaction round (mutual cooperation (CC); cooperation while the interaction partner defected (CD); defection while the interaction partner cooperated (DC) and mutual defection (DD)). These genes can only take values 0 or 1

individual implements substrategy C1; otherwise, the individual implements substrategy C2. Each substrategy (C1, C2 and H) consists of four loci prescribing whether the individual cooperates or defects depending on the outcome of the previous interaction round (i.e., we only allow pure strategies with memory one), and an additional locus prescribing the probability that the individual cooperates in the first interaction round. Individuals occasionally make mistakes in the implementation of their strategy (with probability $\varepsilon = 0.01$).

After all ten repeated interactions finish, individuals reproduce asexually, proportionally to the payoffs they have accumulated ('roulette wheel selection'). New individuals inherit the genes of their parent, with a small chance of mutation ($\mu = 0.001$). To ensure that our results are robust with respect to the specifics of mutation, we ran 50 replicate simulations for each of 100 different matrices determining the probabilities with which the mutation of each strategy produces each other strategy (see Methods for details). We ran all simulations for 10,000 generations, and report

the outcomes (cooperation level, percentage of individuals using heuristics and strategy frequencies) in the last generation.

**Simulation outcomes**. As expected, our results show that uncertainty about the nature of social interactions leads to the evolution of social heuristics (Fig. 2a). Specifically, for high uncertainty ($u > 0.6$), heuristic decision- making strategies were

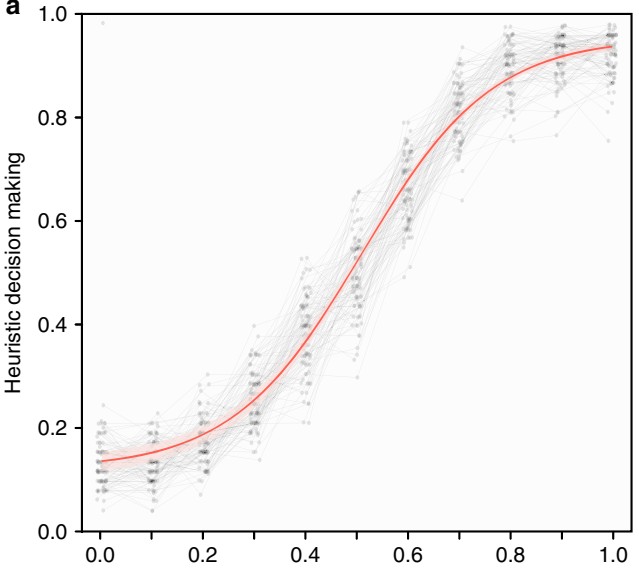

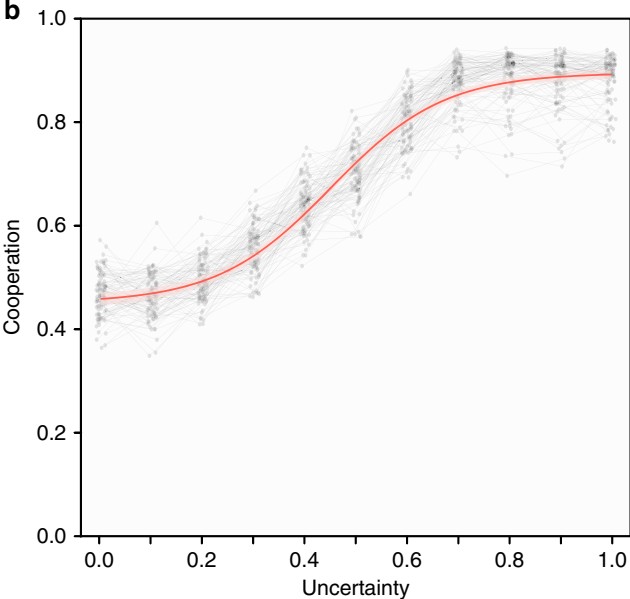

**Fig. 2** Uncertainty about the nature of social interactions leads to the evolution of cooperative heuristics. **a** The fraction of individuals using heuristic strategies and **b** the average cooperation rate at the end of evolutionary simulations that impose varying levels of uncertainty about social interactions. Grey lines show separate results for 100 different random mutation matrices that determine the probability with which a mutation of each strategy gives rise to each other strategy (every point shows an average over 50 replicate simulations per mutation matrix). The red line and shading provide the estimate of the mean and 95% confidence interval of (**a**) heuristic decision making and (**b**) cooperation for the average mutation matrix (modelled as a four-parameter logistic function, see Methods for details)

used by on average 83.3% of individuals at the end of the simulations, whereas this was only the case for 10.3% of individuals under low uncertainty ($u < 0.4$). For intermediate values of uncertainty ($0.4 < u < 0.6$), the probability that evolution leads to a population dominated by heuristic strategies rather than context-dependent strategies was considerable, and gradually increased with uncertainty. Higher uncertainty did not only lead to more heuristic decision making, but also to higher mean levels of cooperation (Fig. 2b). Under high uncertainty, simulations reached an average cooperation level of 0.87, whereas they reached a cooperation level of only 0.46 under low uncertainty.

Why does high uncertainty lead to more cooperation in our simulations? To obtain more insight into this, we investigated which strategies most commonly emerged (Fig. 3). Under low uncertainty, the four most commonly evolved strategies were all context-dependent strategies that combine a tendency to defect when cooperation is relatively costly with a tendency to cooperate if a single act of cooperation is directly beneficial or carries a low cost. As a consequence, populations that mostly consisted of individuals following any of these strategies reached intermediate cooperation levels (between 0.24 and 0.53). Under high uncertainty, the most commonly evolved strategy (by far) was grim, which only cooperates if both interaction partners cooperated in the previous round, and otherwise defects. Because this strategy also evolved a high probability to cooperate in the first round (on average 0.99), an interaction of two individuals employing this strategy likely results in sustained cooperation, until one of the individuals makes a mistake (since the repeated interaction only lasts for ten rounds, this is relatively unlikely). Consequently, a population consisting only of individuals following this strategy reached an average cooperation level of 0.90. The only other heuristic strategy that commonly evolved under high uncertainty was tit for tat with an average initial cooperation probability of 1.00, which led to even higher cooperation levels (0.94). In sum, high uncertainty led to the evolution of heuristic strategies that achieved high cooperation levels when common in the population.

Why does evolution tend to produce the strategies described above? It is not that straightforward to answer this question; the strategy space in our simulations is quite large and the process of social evolution can be highly intricate[23,24]. However, there are some ways in which we can obtain some more insight into why we see these specific strategies emerging. First, we performed a simple version of an invasion analysis in which we assess the fitness of the most commonly evolved heuristic strategy and the most commonly evolved context-dependent strategy against each other for various levels of uncertainty (Supplementary Note 1; Supplementary Fig. 1). This analysis shows that the invasion fitness of the heuristic strategy increases with uncertainty, whereas the opposite is true for the context-dependent strategy. In fact, the heuristic strategy can invade a population of the context-dependent strategy if uncertainty is high enough, whereas neither strategy can invade the other for lower values of uncertainty. Although this gives only a limited picture (only a very small fraction of the strategy space is considered), it does give a rough idea of how uncertainty affects the fitness of the strategies that evolved in our model. Second, we compared our simulation results with the outcomes of benchmark simulations in which we keep the value of $c$ constant (hence, there is no heterogeneity nor uncertainty in social interaction types; Supplementary Note 2). In these simulations, high cooperation levels evolve for values of $c$ as low as $-1.2$ (Supplementary Fig. 2). Also, we observe that evolution produces similar cooperation levels and strategies in a world where individuals always face interactions in which $c = -1$ (i.e., the 'average game' in our original setup) as under maximal uncertainty in our original

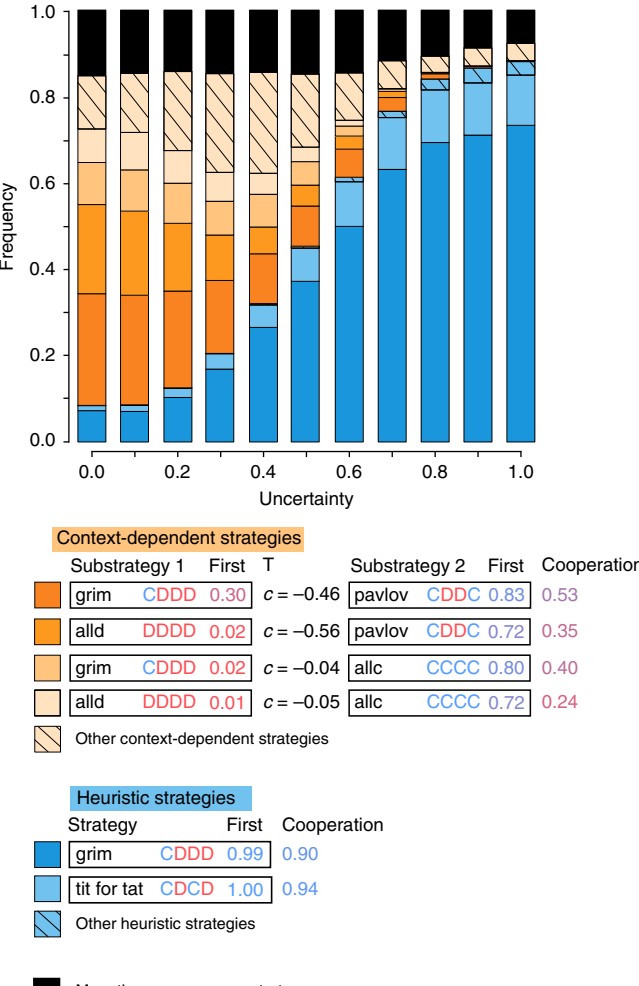

**Fig. 3** The most common context-dependent and heuristic strategies that evolved for various levels of uncertainty about social situations. Each bar shows the fractions of 5000 replicate simulations that were dominated by the strategies shown below (a strategy was considered to dominate if it constituted more than 80% of the population by the end of the evolutionary simulation). The most common context-dependent strategies are shown in orange. These strategies are defined by two substrategies and a threshold (*T*; if the individual perceives *c* to be below this value, substrategy 1 is implemented; otherwise, substrategy 2 is implemented). Each substrategy is defined by five genes, which determine whether the individual cooperates (C) or defects (D) given the outcome of the four possible outcomes of the previous round (Fig. 1), and by a locus that determines the probability that the individual cooperates in the first interaction round. The last column gives the cooperation rate in a population that consists only of individuals implementing the given strategy. The most common heuristic strategies are shown in blue (these implement the same strategy independent of specifics of the social interaction at hand). Black bars show simulation runs in which no single strategy achieved a frequency above 0.8

simulations (Supplementary Fig. 3). This suggests that evolution under maximal uncertainty produces strategies that are tuned to the average social interaction context that individuals face.

It has previously been shown that the specifics of the mutation process can dramatically alter the outcome of social evolution[23,25]. For this reason, we replicated our simulations 100 times, every time randomising the probabilities with which each strategy gives rise to each other strategy in the event of a mutation. Even though this did lead to some differences in the frequencies with which the different strategies emerged as the outcome of evolution, the overall picture remained virtually the same regardless of the mutation probabilities, with uncertainty about the nature of social interactions consistently leading to the evolution of cooperative heuristics (see grey lines in Fig. 2).

## Discussion

Our simulations show that social heuristics evolve when individuals face many different social interaction types in their lives and have uncertainty about the nature of the interactions they face. The social heuristics in our model were likely to evolve when uncertainty was intermediate to high, and were associated with higher levels of cooperation than the more sophisticated strategies that evolved under low uncertainty. These outcomes were robust to the specifics of mutation—even though different mutation implementations could lead to the evolution of different strategies (especially under low uncertainty), the overall picture with respect to the prevalence of heuristic strategies and cooperation levels remained the same.

Our results provide a potential explanation for why humans are often observed to cooperate in circumstances where standard theory would not predict them to do so[26–28]. Just like in the real world, many instances of cooperative behaviour that emerge in our model seem maladaptive when considered in isolation. The most common social heuristic that evolves in our model cooperates even if the cost of cooperation exceeds the benefit to the interaction partner—a type of extremely altruistic behaviour that standard models of social evolution would never predict to evolve[29,30]. Indeed, as we show in our simulations in which the social interaction type remains fixed, evolution in isolated social contexts does not produce cooperative outcomes in that part of the parameter range (Supplementary Fig. 2). Our model shows that such seemingly suboptimal behaviour can in fact be adaptive when the heterogeneity and uncertainty of the world that it has evolved in are taken into account. If there is uncertainty, strategies that generalise well across situations, even if they perform poorly in some cases, have an advantage over more sophisticated strategies that risk overfitting their behaviour to the specific social interaction type they believe they are facing (see also the invasion analysis in Supplementary Note 1).

In our model, low uncertainty leads to strategies that only cooperate if a single act of cooperation is directly beneficial or carries a small cost (i.e., if *c* exceeds either −0.5 or 0; Fig. 3). Depending on exactly which context-dependent strategy evolves, cooperation levels in these simulations range from 0.24 to 0.53 (on average 0.45). These outcomes contrast with the evolution of strategies in the absence of any heterogeneity in social interaction contexts (see Supplementary Note 2). There, high cooperation levels evolve even if costs of cooperation are moderate (as long as *c* > −1.2), leading to higher overall cooperation levels over the entire parameter range (0.55). This shows that even in the absence of uncertainty, heterogeneity in social interaction contexts alone can already lead to evolutionary outcomes that are at odds with expectations based on evolution in isolated social contexts. In sum, our simulations provide proof of principle that both heterogeneity in social interactions and uncertainty about social interactions can strongly affect overall cooperation levels, and lead to behaviours that appear to be suboptimal when considered in isolation.

We have attempted to design a model that is sufficiently simple to interpret the outcomes, but not so simple that the outcomes become trivial. However, as every modeller knows, simplicity comes at a cost to realism. We have only considered heterogeneity in a single parameter of the social interaction types (*c*) and have chosen to consider a range of this parameter that spans social

contexts that (in isolation) should be expected to lead to the evolution of full cooperation (if $c > 0$) and full defection (if $c < -2$; cf. Supplementary Fig. 2). In reality, social interaction types vary in more than a single dimension, and the effect of uncertainty on cooperation levels may vary with the parameters that are considered heterogeneous, and the range that they are varied over. Similarly, to make sure that evolution can still efficiently search the strategy space, we have chosen to confine it to include only pure substrategies with memory of a single interaction round. Confining the strategy space prevents high path dependence in the evolutionary outcomes, which would make it very hard to draw general conclusions. However, many other choices with regard to the strategy space can be reasonable—they may each influence the effect of uncertainty on strategy evolution and cooperation differently. In sum, our model shows that heterogeneity in and uncertainty about social interaction types can strongly affect the evolution of cooperation, but the direction and magnitude of this effect will likely depend on the details of the social environment and the cognitive architecture. Hence, empirical work to inform these model assumptions is essential to make further progress in understanding the effect of uncertainty on cooperation.

## Methods

**Overall simulation setup**. Individuals in the first generation of each simulation were constructed by initialising the gene at locus $S$ to be 0 (context-dependent) or 1 (heuristic) with equal probability, initialising the probability of cooperating on the first move for all three substrategies by randomly drawing a number from a uniform distribution between 0 and 1, initialising the conditional parts of their three substrategies by randomly assigning one of 16 possible strategies with equal probability, initialising the threshold gene $T$ at 0 and initialising their fitness at a baseline fitness of 100.

We ran all simulations for 10,000 generations, and reported the outcomes (cooperation level and percentage of individuals using heuristics) in the last generation. We ran separate simulations for 11 levels of uncertainty ($u$), ranging from 0 to 1 in steps of 0.1, and 100 randomly generated mutation matrices (see below). For each of the resulting 1,100 combinations of the degree of uncertainty and mutation matrix, we ran 50 replicate simulations. All simulation codes were written in C++.

**Mutation**. Mutation probabilities were the same for all loci ($\mu = 0.001$). In the event of a mutation at a locus $S$ (which determines whether the individual uses a context-dependent or a heuristic strategy), the gene mutated from 0 to 1 or vice versa. In the event of a mutation at locus $T$ or at any of the loci determining the first moves of the substrategies, the value of the new gene was equal to the value of the parent gene added to a value drawn from a normal distribution with mean 0 and standard deviation 0.1.

The mutation of the conditional parts of the substrategies (indicated by subscripts 1–4 in Fig. 1c) proceeded according to a predefined mutation matrix. This matrix specified all probabilities that each possible substrategy mutated into each possible other substrategy (there are 16 possible substrategies; Supplementary Table 1). We ran all simulations for 100 randomly generated mutation matrices. These mutation matrices were generated in such a way that each substrategy could only mutate to four other substrategies (with probability 0.25). These four substrategies were randomly drawn from all 15 other substrategies with equal probability (it was not possible for a mutation to generate the same strategy). We used this way of generating mutation matrices to ensure that we covered a large diversity of constraints on evolution, allowing us to get a comprehensive idea of the robustness of our results. This is in contrast to a strategy of generating mutation maps where all entries of the mutation matrix are drawn from a uniform distribution (and later normalised), which would result in non-zero probabilities of mutation from each substrategy to each other substrategy and which would impose only relatively mild constraints on the evolutionary process.

**Statistical model**. We modelled the percentage of heuristic strategies (Fig. 2a) and the level of cooperation (Fig. 2b) according to a four-parameter logistic function of uncertainty, according to the following formula:

$$x = \frac{y_{\min} + (y_{\max} - y_{\min})}{1 + e^{\frac{u_{\mid} - u}{\beta}}} \tag{1}$$

where $y_{\min}$ and $y_{\max}$ indicate the lower and the upper asymptotes of the logistic function, $u_{\mid}$ indicates its inflection point and $\beta$ indicates its slope. We constructed a model in which we first included all four of these parameters as well as

uncertainty ($u$) as fixed factors and the mutation matrix as a random factor and then simplified this full model on the basis of AIC. In the final models, both of percentage of heuristic strategies and of cooperation level, all parameters except the slope ($\beta$) varied with respect to the mutation matrix. The confidence intervals depicted in Fig. 2 in the main text were obtained by sampling from the multivariate distribution of parameter estimates of only the fixed effects of the model. These analyses were performed using the 'nlme' package in R.

**Invasion analysis**. We conducted a simple version of an invasion analysis, based on simulations, of the two most common strategies that emerged in the model. Specifically, we ran simulations to ascertain the payoffs that the most common heuristic strategy (grim) and the most common context-dependent strategy (grim/pavlov— Fig. 3) obtain against each other and against themselves. From this, we calculated invasion fitness, which is defined as the fitness that a mutant obtains in a resident population, divided by the fitness that the resident obtains against itself. If this number exceeds 1.0, the mutant strategy can invade. The results of this analysis are reported in Supplementary Note 1.

**Simulations for fixed values of $c$**. We conducted simulations of a version of our model without heterogeneity in social interaction types (nor uncertainty about the interaction context). In these simulations, because individuals are always interacting in the same interaction type, they can only have a simple strategy consisting of a continuous first-move locus and four Boolean loci that determine behaviour in later rounds of each repeated interaction (corresponding to a 'substrategy' in the main model). Otherwise, the details of the simulations (number of interactions, interaction length and mutation probabilities) are all the same as in the original model. We ran separate simulations for a number of different values of $c$. The results of these simulations are reported in Supplementary Note 2.

**Code availability**. The simulation code used in this study has been made available at GitHub (https://github.com/pvdberg1/uncertainty_cooperation_evolution).

**Data availability**. All data are available from the corresponding author upon request.

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

## Acknowledgements

P.v.d.B. is supported by a Rubicon fellowship (File No. 019.2015.2.310.041) of the Dutch Science Foundation (NWO) and a postdoctoral fellowship (App. No. G093618N) of Research Foundation—Flanders (FWO).

## Author contributions

P.v.d.B. designed research, ran simulations, analysed data and wrote the manuscript. T.W. designed research and wrote the manuscript.

## Additional information

**Competing interests:** The authors declare no competing interests.

