## [Peer Review File · Nature Communications]

Reviewers' comments:

Reviewer #1 (Remarks to the Author):

The major claim of this paper is that uncertainty in social interactions leads to the evolution of social heuristics, i.e, strategies that are applied across all situations encountered, rather than conditional strategies (,those that make an assessment of a given situation, and act differently depending on the basis of that assessment).

This is an important paper because it moves us away from the highly artificial structure of many evolutionary games/models of cooperation. Such games often assume the players have perfect information, and that they know the nature of the game they are playing. Real-life isn't like that: most of the time we have to figure out the rules of the game as we go along, and this often involves a high level of uncertainty. The question is then whether we really do figure out the precise nature of the game we're playing or whether we use rules-of-thumb that are good enough to let us get by. By incorporating uncertainty into their model, Van den Berg and Wenseleers show that cooperative social heuristics are indeed more likely to evolve as uncertainty increases. They also show that such cooperative strategies are more common and more cooperative than they 'should' be, if they were taken in isolation with no appreciation of the social environment.

These findings will undoubtedly be of interest to others in the community, and would have widespread application in terms of both the development of further theory and empirical studies. For example, I can see how this work has implications for the personality-plasticity literature: previous theoretical work by Mathot et al. (2012) suggests that personality traits and how these relate to behavioural plasticity may be underpinned by variation in organisms' response to environmental uncertainty. Showing that uncertainty in social interactions drives the evolution of heuristics -- which one can conceptualize as personality traits -- adds another dimension to this work, and the two could be combined very profitably.

The model and analyses are clear and well explained, and I would imagine a competent theoretician could reproduce the work from the details provided (I am not a competent modeler is has to be said, so you might need a further opinion on that! But given that I'm not gifted in this way, the fact that I followed everything and appreciated the logic suggests it is very well done).

I have a couple of minor comments:

1. While I appreciate that most methodological detail these days is doomed to reside only in supplementary material, due to word limits and other aspects of journal policy, I do think this is a mistake. Understanding of the current paper would be enhanced by more detail in the main text. For example, a description of the strategies investigated and how they work would be really handy, so a reader can think more deeply about why particular social heuristics do better than others.

2. This is probably a stupid point, but I would have liked more explanation of exactly why a strategy like grim does so well as a social heuristic--why does it generalise so effectively

across all situations? -- and why it is that high levels of cooperation evolve under uncertainty? For example, is it because conditional strategies are really punished under uncertainty, which means social heuristics flourish, or do social heuristics have some absolute advantage when conditions are uncertain or is that a stupid question? I suppose what I'm also saying here is I would have been interested in more details on the actual evolutionary trajectories seen, or some other way of showing how and why social heuristics start to gain an advantage and why as uncertainty increases. Or is this irrelevant to the question at hand here? i.e., it doesn't really matter what the strategies are as such, all that matter is that heuristics do better than conditional strategies?

Reviewer #2 (Remarks to the Author):

This is an interesting manuscript on the impact of uncertainty on the evolution of social behaviour. I think the impact of uncertainty is indeed of fundamental importance for evolution in general, and the authors point to some relevant references. Unfortunately they seem to miss one highly relevant paper that presents results congruent with their own:

McNamara et al. (2004) Variation in behaviour promotes cooperation in the Prisoner's Dilemma game. *Nature* 428(6984):745-8

There the authors show that fixed length iterated PDs need not converge to defection on the first round, under the standard logic, provided there is variation in the population as to the number of rounds individuals will cooperate for.

The authors need to cite this paper and relate their results to it, since it has very clear priority. The impact of the present work might be considered to be reduced by this precedent. I don't like impact considerations, but understand this might be a criterion for *Nature Comms*.

Other minor comments:

In reading the text, when u and cooperation level are introduced it would be helpful to briefly explain how they work and especially what their intervals are, even if just by reference to the figures (e.g.: $u > 0.6$ (u in interval 0 to 1; see fig 1)

line 74: 'when cooperation is directly beneficial or carries a low cost' - here I think care is needed since, as the authors will know, there had been a tendency to confuse direct stage-game costs (i.e. per round) and overall expected costs from a complete iterated interaction, with unfortunate consequences. Since I assume evolution takes place in an unstructured population, I suppose evolution should only favour cooperation when net beneficial, on average. Against this, the authors briefly discuss 'hyperaltruism' - this is unexpected and deserves further discussion, as it seems to run counter to the logic just outlined.

fig 1: the choice of sign for c is non-standard and could confuse; usually $c < 0$ is taken to be directly *beneficial* rather than costly behaviour (as described under the convention of

Hamilton's rule); here it appears c 's sign is opposite, so $c > 0$ is directly beneficial

fig 1. the uncertainty function clearly induces bias, as well as increasing variance in estimates of cooperativeness - any non-zero value of u will lead to a mean perceived cooperativeness *below* the true environmental value. This is poor statistical practice; what happens if this assumption is changed? Are the results qualitatively changed? As it stands, the authors are vulnerable to a claim that they have confounded bias and uncertainty.

fig 3. the labels 'heuristic' and 'conditional' seem not particularly discriminatory (TFT is arguably both heuristic and conditional, for example; I'm sure it has been described as both repeatedly in the literature) and the way they were applied was opposite to what I felt natural. Perhaps some other less ambiguous labels can be found.

Reviewer #3 (Remarks to the Author):

Please see attached pdf file.

Reviewer #4 (Remarks to the Author):

In this paper the authors study, using a simulation model, the evolution of cooperative behaviour when there are uncertainties in the types of social interactions that can arise. The payoff structures vary in the different types of interactions considered, ranging from situations where cooperation is always the best action for the focal individual (regardless of the behaviour of the interaction partner), to versions of the Prisoner's Dilemma, to situations where it never pays to cooperate. The authors attempt to study how the degree of uncertainty that individuals have about the nature of the social interactions affects: (1) the evolution of heuristic decision making, and (2) the level of cooperation in the population. They find in their simulations that uncertainty can lead to the evolution of simple heuristic strategies that disregard the available information about the nature of the social interaction, and in addition, that these heuristic strategies are more cooperative than the strategies that evolve when individuals are better able to assess social interactions.

Studying the effects of uncertainties in the types of social interactions on the evolution of cooperation is a potentially interesting topic, however, the results found in the current paper are rather trivial. It is essentially obvious, due to the way the authors have defined their model, that increased uncertainty will result in increased use of heuristic strategies. Moreover, it is clear that the level of cooperation may increase, since the authors are allowing for situations in which cooperation is always the best action. The more interesting question would be how the frequency of cooperation depends on the various types of payoff structures that are encountered, and the authors have not analyzed this important question in any detail. An additional, and potentially serious, shortcoming of the approach adopted here is that only deterministic strategies are considered. This assumption is at odds with the

authors' notion that the individuals are interacting in an uncertain world — in such a world errors due to the mis-implementation or mis-reading of strategies would be expected to be common, and thus, the more natural framework to use would be that of stochastic strategies. Some of the key results that the authors find in their simulations may be quite different with stochastic strategies. For example, GRIM was found to do well in the simulations (since it cooperates with itself), however, this would almost certainly change if the strategies allowed for errors, since an occasional mistake would then result in long stretches of mutual defection. Finally, the format of the paper is unsuitable for Nature Communications — the body of the paper is very brief, with most of the important details relegated to the Supplementary Information.

Review of:

“Uncertainty about social interactions leads to the evolution of cooperative heuristics”

This is a paper on an interesting topic – the evolution of cooperative heuristics. I associate this idea with David Rand’s group (e.g. Rand 2014), which they cite. The idea is that people play games with each other, but are uncertain about what games they are playing. They, therefore, develop heuristics that plays some average of the games. I think this has the potential to be an interesting model and publication.

However, I have some concerns about how the model is implemented and make various suggestions for improving it and conducting simpler version of the model to put their findings into context.

I especially think that the authors need to spend some time explaining their key result. Heuristics evolve, but they do not explain *why* heuristics evolve. I have my suspicions, but the authors need to do a little more work to help the reader understand what is going on with their model.

I have some concerns about implementation of the model that may be giving misleading results. I am a modeler and understand that there are a vast number of ways to implement a model and decisions must be made. I get annoyed when reviewers suggest what I see as unnecessary complications or changes to one of my models. Therefore, when suggesting something different or more complicated, I will do my best to explain why I think it is necessary or important. The authors may find a different way to address my concerns that I did not consider.

I recommend a major revision to include additional simulations.

Major comments:

1. Since it is the main finding of the paper, the authors really need to explain why heuristic strategies evolve in their simulations. By what mechanism do they out-compete conditional strategies? What is the selective advantage of using a heuristic strategy over a conditional strategy? Since this is the whole point of the paper, the authors really need to explain it.

Ideally the authors would do this as analytically or numerically as possible. For a given set of parameter combinations, what is the payoff to a conditional strategy and a heuristic strategy?

2. I suspect that the real reason that heuristic strategies evolve is because the authors implement an external cost to using a conditional strategy. This is somewhat buried in the appendix: “Individuals with a conditional strategy paid a fitness cost of 1%.” This important fact needs to be moved into the main body of the paper. I think it should also be the authors’ working hypothesis as to why heuristic strategies evolve unless they have a better explanation.

This is not that exciting of an answer, though. Imposing an external cost on a strategy is an easy way to get it to evolve less often. The authors should be able to tell how much this cost matters either analytically or numerically.

The authors need to move this information to the main body of the paper and discuss whether this is what is driving the evolution of heuristic strategies.

3. The authors describe this external cost in the appendix. “This small cost for using a conditional strategy was implemented for practical reasons: if there would be no such penalty, there is the possibility that conditional strategies evolve to be *de facto* heuristic (if the threshold gets very close to -3 or 1), obscuring an appropriate analysis of the degree to which heuristics are used.”

I do not think this is the most parsimonious solution to this problem. Since this is a simulation model, the authors have access to the internal states of their agents. They should be able to tell if the threshold gets very close to -3 or 1 and just report it.

Similarly, a *de facto* heuristic strategy can evolve if both of the conditional strategies are the same. It should be easy for the authors to check for this in their code. The authors, at a minimum, should run the simulations without the external cost to conditional strategies and see how it changes their results. I suspect that the heuristic strategy would have a hard time evolving without penalizing the alternative.

4. Another option would be for the authors to just be up-front about a cognitive cost driving heuristics, if that is what is driving their results. This might not be as interesting as getting their results without a cognitive cost, but it is still interesting.

5. The authors need to explain this better “To avoid computational problems stemming from negative payoffs, a constant was added to each of the payoffs so that the lowest possible payoff was equal to zero.” Does this mean that a 3 was added to all elements of the payoff matrix when $c = -3$ and nothing is added to the payoff matrix when $c = 1$? Or is 3 just added to all of the elements of all of the matrices? Either way, this needs to be more explicit than the vague statement in the appendix. The authors should just modify their discussion of the payoff matrices on Figure 1a with the actual values to include this bit that was added.

I am not very comfortable with relative payoffs this close to zero. This implies that social interactions are all that matter to fitness. Normally, in evolutionary game theory, we implement a “baseline fitness” to take care of negative numbers and zeros. Baseline fitness is also useful because evolution maximizes the geometric mean fitness and thus really tries to avoid zero fitness. This can lead to all sorts of asymmetries with the specific values in the payoff matrix. Generally, the higher the baseline fitness the better the estimate of the direction of selection, but the slower selection occurs.

In summary, to avoid negative numbers, the authors should just shift the payoff matrices to all positive numbers. I also think that they should implement a baseline fitness. They would need to try different values to see what gives them a good balance, but something like a baseline fitness of 50-100 would probably work (with expected payoffs from the games running from 0 to 60 - ~10 rounds with scores of 0 to 6 per round).

6. I am also uncomfortable with the way that the authors implement uncertainty.

$$c_p = c * (1 - \mu) + \varepsilon * \mu$$

This results in the player's observation of the cost of cooperating being uniformly distributed and not even centered on the observed value. I do not think there is any implementation of signal-detection theory that would use this distribution. This might really change the results by because it not only gives the player a noisy signal, but a biased and mostly uninformative signal (Figure 1).

Figure 1. Current distribution from which the observed value of c (c_p) is drawn when $c = 0$ (red dashed line) and $\mu = 0.5$. The distribution is uniform and not centered on c . This might make the signal worse than it should be and therefore, heuristic strategies are likely less useful than they would be with a more realistic signal distribution.

It is much more reasonable if the mode of c_p was c with likelihood decreasing away from c_p . The authors should change to this type of probability distribution to make sure that their results are not due to misleading observations.

The authors could implement this by setting c_p as a random number drawn from a beta distribution where α is the uncertainty parameter (from 1 to ∞) and $\beta = \frac{4(\alpha-1)}{c+3} + 2 - \alpha$ that is

then multiplied by 4 and added to -3. All of these 4s and 3s are in there to keep c_p on the interval from -3 to 1 (see Figures 2-4).

Figures 2-4. An implementation using the beta distribution as described in this review. Now the modes are at the actual value of c (red dashed line) and the likelihood of the signal falls as one gets away from c . These are for $\alpha = 10, 100, 1000$. $\alpha = 1$ gives a uniform distribution over the whole range (similar to $u = 0$ in the submitted version).

I've appended the code for generating these figures at the end of this review. This would aid the authors in implementing a more reasonable error function.

7. The authors need to show how their results differ from null models. This can be done in the appendix and referenced in the main paper.

The first null model would be one where there is no variation in the payoff matrix. The reader should be able to compare the results of the more complicated models to the simpler models. They should run their model with constant game matrices at various values of c . Half unit intervals (i.e., $c = -3, -2.5, -2, -1.5, -1, -0.5, 0, 0.5, 1$) would be sufficient. What strategies evolve under each of the the steady-state conditions? How much cooperation is there?

This will let the reader see to what extent the results in, say, Figure 2b and 3 are an average of the steady-state games. It also will give a robustness check.

There would not need to be any heuristic strategies in this model since there is no uncertainty over the game type.

8. Another simpler model that the authors should report (in the appendix, referenced in the paper) would be one where players only use strategies of pure cooperation or pure defection. This would, again, give readers the ability to see whether the strategies are just averaging over

the games. The authors would implement both conditional and heuristic strategies in this version of the paper. This would give the readers a baseline with which to compare the more complicated strategies.

I actually think the paper would be more convincing if they just used these simple strategies instead of introducing the more complicated repeated game structure. It would be a more tractable implementation of their idea.

Another advantage of this simpler version of their model is that the authors should be able to give an analytic analysis of the expected payoffs for cooperating or defecting under various values of c_p with given values of μ . They could even plot these and the reader could compare these to their results from the more complicated model.

9. If the authors want to have more a more complicated strategy space, I am uncomfortable with them using strategies that only remember the previous interaction. The reason is that the invasion dynamics for repeated games are complicate, these complications only arise with forgiving strategies of memory two or greater, and focusing only on memory one strategies can be misleading. This is a bigger deal because the authors' conclusion is that their "model provides insight into why individuals often behave more cooperatively than standard evolutionary and economic theory would predict."

The reasons for these complications are the folk theorem of repeated games and findings that there is no evolutionarily stable strategy in the iterated prisoner's dilemma. For example:

Boyd, Robert, and Jeffrey P. Lorberbaum. "No pure strategy is evolutionarily stable in the repeated Prisoner's Dilemma game." *Nature* 327.6117 (1987): 58-59.

As an example of where overly simple strategies can lead a paper wrong, the authors should look at a paper they cite:

Delton, Andrew W., et al. "Evolution of direct reciprocity under uncertainty can explain human generosity in one-shot encounters." *Proceedings of the National Academy of Sciences* 108.32 (2011): 13335-13340.

A critique pointed out that adding slightly more complicated strategies could destroy cooperation in the exact same model:

Zefferman, Matthew R. "Direct reciprocity under uncertainty does not explain one-shot cooperation, but demonstrates the benefits of a norm psychology." *Evolution and Human Behavior* 35.5 (2014): 358-367.

Plus additional commentary of interest:

Delton, Andrew W., and Max M. Krasnow. "An independent replication that the evolution of direct reciprocity under uncertainty explains one-shot cooperation: Commentary on Zefferman." *Evolution and Human Behavior* 35.6 (2014): 547-48.

Zefferman, Matthew R. "When repentance and forgiveness are possible, direct reciprocity does not explain one-shot generosity, even under arbitrarily high levels of uncertainty." *Evolution and Human Behavior* 35.6 (2014): 548-49.

I worry that the authors are introducing a similar problem in their model as the Delton and Krasnow paper. To make a stronger claim about the implications of their model to human cooperation, the authors should implement their model with strategies of at least memory two.

Note: I know that it may seem contradictory to both use simpler and more complicated strategies. Specifically, I am suggesting that simpler strategies would make their findings about the evolution of heuristics more tractable. More complicated strategies are needed to make their findings about the evolution of cooperation more believable.

10. The authors should provide their code and results in a publically accessible data and code archive, such as GitHub, BitBucket or the Open Science Framework. The authors should report, in the paper, where their code and data is or will be stored.

More minor comments:

11. The set-up for their model is very similar to "multiple games theory" papers by Bednar and co-authors. In these games, individuals play iterated games with uncertain payoff matrices. They talk about this in terms of heuristics and there is even an experimental literature on this. The authors could start with the below papers to familiarize themselves with this literature. Should acknowledge and address similarities in the models and ideas in these papers and their submission.

Bednar, Jenna, and Scott Page. "Can game (s) theory explain culture? The emergence of cultural behavior within multiple games." *Rationality and Society* 19.1 (2007): 65-97.

Bednar, Jenna, et al. "Behavioral spillovers and cognitive load in multiple games: An experimental study." *Games and Economic Behavior* 74.1 (2012): 12-31.

Grimm, Veronika, and Friederike Mengel. "An experiment on learning in a multiple games environment." *Journal of Economic Theory* 147.6 (2012): 2220-2259.

12. The authors should justify randomizing the number of interactions. Why not make things simpler by keeping them constant at 10? Since they are only using strategies with memory one (or two), there is no danger of backwards induction.

Figure generating R code:

```
# Uniform distribution used in paper
```

```
c = 0
```

```
u = .5
```

```
cp_distribution = c * (1-u) + runif(10000000, min = -3, max = 1)*u
```

```
hist(cp_distribution, breaks=100, xlim=c(-3,1))
```

```
abline(v=c, lty=2, col="red", lwd=3)
```

```
# Suggested beta distribution
```

```
c= 0
```

```
mode = (c + 3)/4
```

```
alpha = 1
```

```
beta = (alpha - 1)/mode + 2 - alpha
```

```
cp_distribution = 4*rbeta(10000000, shape1=alpha, shape2=beta, ncp = 0)-3
```

```
hist(cp_distribution, breaks=100, xlim=c(-3,1))
```

```
abline(v=c, lty=2, col="red", lwd=3)
```

'Uncertainty about social interactions leads to the evolution of social heuristics'

Manuscript by: Van den Berg and Wenseleers

Response to reviewers

This document contains a point-by-point description of how we revised our manuscript in response to the reviewers' concerns. We would first like to take the opportunity to thank the reviewers for their comments – they were highly valuable for improving this manuscript. Reviewer remarks will be printed in black, and a concise description of our response will be printed in blue.

Reviewer #1

The major claim of this paper is that uncertainty in social interactions leads to the evolution of social heuristics, i.e. strategies that are applied across all situations encountered, rather than conditional strategies (those that make an assessment of a given situation, and act differently depending on the basis of that assessment).

This is an important paper because it moves us away from the highly artificial structure of many evolutionary games/models of cooperation. Such games often assume the players have perfect information, and that they know the nature of the game they are playing. Real-life isn't like that: most of the time we have to figure out the rules of the game as we go along, and this often involves a high level of uncertainty. The question is then whether we really do figure out the precise nature of the game we're playing or whether we use rules-of-thumb that are good enough to let us get by. By incorporating uncertainty into their model, Van den Berg and Wenseleers show that cooperative social heuristics are indeed more likely to evolve as uncertainty increases. They also show that such cooperative strategies are more common and more cooperative than they 'should' be, if they were taken in isolation with no appreciation of the social environment.

These findings will undoubtedly be of interest to others in the community, and would have widespread application in terms of both the development of further theory and empirical studies. For example, I can see how this work has implications for the personality-plasticity literature: previous theoretical work by Mathot et al. (2012) suggests that personality traits and how these relate to behavioural plasticity may be underpinned by variation in organisms' response to environmental uncertainty. Showing that uncertainty in social interactions drives the evolution of heuristics – which one can conceptualize as personality traits -- adds another dimension to this work, and the two could be combined very profitably.

The model and analyses are clear and well explained, and I would imagine a competent theoretician could reproduce the work from the details provided (I am not a competent modeler as has to be said, so you might need a further opinion on that! But given that I'm not gifted in this way, the fact that I followed everything and appreciated the logic suggests it is very well done).

I have a couple of minor comments:

1. While I appreciate that most methodological detail these days is doomed to reside only in supplementary material, due to word limits and other aspects of journal policy, I do think this is a mistake. Understanding of the current paper would be enhanced by more detail in the main text. For example, a description of the strategies investigated and how they work would be really handy, so a reader can think more deeply about why particular social heuristics do better than others.

This is a good point. The manuscript was initially drafted for *Nature* and subsequently transferred to *Nature Communications* from there. This is the reason that it was in a very short format and that all methodological details were provided in the Supplementary Information. For the current version, we

have moved the entire model description to the main text. We now also introduce most of the model in the Results section (since the journal format does not allow for a section between Introduction and Results), and have left more technical model details to the Methods sections at the end of the paper.

2. This is probably a stupid point, but I would have liked more explanation of exactly why a strategy like grim does so well as a social heuristic—why does it generalise so effectively across all situations? -- and why it is that high levels of cooperation evolve under uncertainty? For example, is it because conditional strategies are really punished under uncertainty, which means social heuristics flourish, or do social heuristics have some absolute advantage when conditions are uncertain or is that a stupid question? I suppose what I'm also saying here is I would have been interested in more details on the actual evolutionary trajectories seen, or some other way of showing how and why social heuristics start to gain an advantage and why as uncertainty increases. Or is this irrelevant to the question at hand here? i.e., it doesn't really matter what the strategies are as such, all that matter is that heuristics do better than conditional strategies?

This is an excellent question, although it is not necessarily easy to answer. The strategy space in our simulations is quite large, with 17 loci, four of which are continuous (and can therefore take any value in their range). Because of this, it is impossible to do a mathematical analysis to identify the evolutionarily stable strategies under the various uncertainty regimes. Also, because of the complexity associated with social evolution, it is not always straightforward to come to a verbal explanation of why we see the patterns that arise in our simulations – this is partly why we run simulations in the first place. However, there are some ways to get a better understanding why we get the strategies that we do, and we have significantly expanded on this in the manuscript, with a new paragraph in the Results section, two new paragraphs in the Discussion section, and two sections in the Supplementary Information.

First, we have analyzed the performance of the most commonly evolved heuristic and conditional strategies against each other, under various levels of uncertainty. This gives us a rough idea of how likely both strategies are to emerge as dominant, depending on uncertainty (although it is far from a complete picture, because the strategy space is infinitely larger than the two strategies considered here). Interestingly, in most of the uncertainty range, neither strategy is vulnerable to invasion of the other strategy. However, it is clear that the invasion fitness of the heuristic strategy increases with uncertainty, and the invasion fitness of the conditional strategy decreases with uncertainty, which affects invasion chances in a finite population. We present these results in the Supplementary Information (section 3) and discuss them in the Results section (lines 139–150) as well as (briefly) in the Discussion section (lines 185–188).

Second, we have run a large amount of 'benchmark' simulations of a version of the model in which there is no heterogeneity in the social situations individuals face. What we observe is that strategy evolution under complete uncertainty is similar to strategy evolution in the 'average game', so in a world where individuals always face the interaction characterized by $c = -1$. Although the results are not entirely the same, this suggests that complete uncertainty leads evolution to produce strategies for the average game. We present these results in the Supplementary Information (section 4), and discuss them in the Results section (lines 151–158) as well as in the Discussion section (lines 180–183 and 189–200).

Reviewer #2

This is an interesting manuscript on the impact of uncertainty on the evolution of social behaviour. I think the impact of uncertainty is indeed of fundamental importance for evolution in general, and the authors point to some relevant references. Unfortunately they seem to miss one highly relevant paper that presents results congruent with their own:

McNamara et al. (2004) Variation in behaviour promotes cooperation in the Prisoner's Dilemma game. *Nature* 428(6984):745-8

There the authors show that fixed length iterated PDs need not converge to defection on the first round, under the standard logic, provided there is variation in the population as to the number of rounds individuals will cooperate for.

The authors need to cite this paper and relate their results to it, since it has very clear priority. The impact of the present work might be considered to be reduced by this precedent. I don't like impact considerations, but understand this might be a criterion for Nature Comms.

We thank the reviewer for directing us to this excellent paper. As the reviewer mentions, this paper shows how a sufficient amount of variation in strategies (hence, uncertainty about the strategy of the interaction partner) can lead to the evolution of much higher cooperation levels. This is certainly relevant, which is why we now cite this paper (line 37), but the mechanism by which increased cooperation evolves in this paper is completely different from the mechanism by which it evolves in our model. In our model, it is not uncertainty about the strategy of the interaction partner (induced by an increase in mutation of strategies), but the uncertainty about the parametrization of the interaction context itself that affects the evolution of cooperation. So we believe that the relevance of this paper for our manuscript is in providing a general context of how various types of uncertainty can affect the evolution of cooperation (and this is how we have cited it).

Other minor comments:

In reading the text, when u and cooperation level are introduced it would be helpful to briefly explain how they work and especially what their intervals are, even if just by reference to the figures (e.g.: $u > 0.6$ (u in interval 0 to 1; see fig 1)

We have significantly expanded the model description in the main text (rather than leaving this to the Supplementary Information). Most of it is now in the Results section, and the more technical parts are in the Methods section at the end. In the 'The Model' subsection of the Results section, we now clearly mention the minimum and maximum levels of uncertainty that are possible in our model (lines 84–88). This means that the reader now has a much clearer idea of the implementation (and interval) of the uncertainty range before the results are discussed.

line 74: 'when cooperation is directly beneficial or carries a low cost' - here I think care is needed since, as the authors will know, there had been a tendency to confuse direct stage-game costs (i.e. per round) and overall expected costs from a complete iterated interaction, with unfortunate consequences. Since I assume evolution takes place in an unstructured population, I suppose evolution should only favour cooperation when net beneficial, on average. Against this, the authors briefly discuss 'hyperaltruism' - this is unexpected and deserves further discussion, as it seems to run counter to the logic just outlined.

The reviewer is correct that we were a bit careless with our formulations/terminology here. We now reworded this to 'if a *single act* of cooperation is directly beneficial or carries a low cost' (lines 123–126). We also reworded 'hyperaltruism' to 'extremely altruistic behaviour' (lines 178–180).

fig 1: the choice of sign for c is non-standard and could confuse; usually $c < 0$ is taken to be directly *beneficial* rather than costly behaviour (as described under the convention of Hamilton's rule); here it appears c 's sign is opposite, so $c > 0$ is directly beneficial

After careful consideration, we have decided not to change the sign of c . We do agree with the reviewer that some confusion could arise in this respect, because some readers may be used to interpreting c directly as a cost. However, making the opposite choice could be equally confusing to readers who do not have this immediate connotation – they are likely to associate a positive number with a benefit and a negative number with a cost. Another option could be to choose a different symbol for this parameter, but we ended up deciding against this, because we think it is more intuitive if the symbol is directly related to its meaning (c = consequence of cooperation, u = uncertainty). However, to avoid confusion, we have now reworded the introduction of this parameter to make it explicitly clear that a positive value of c denotes a benefit, and that a negative value denotes a cost

(lines 75–79). Together with the visually explicit Figure 1a, we think this should keep confusion about this parameter to a minimum.

fig 1. the uncertainty function clearly induces bias, as well as increasing variance in estimates of cooperativeness - any non-zero value of u will lead to a mean perceived cooperativeness *below* the true environmental value. This is poor statistical practice; what happens if this assumption is changed? Are the results qualitatively changed? As it stands, the authors are vulnerable to a claim that they have confounded bias and uncertainty.

It is indeed true that the uncertainty function in our original model introduced bias. At the suggestion of Reviewer 3, we have now changed the implementation of uncertainty to a beta distribution around the actual value, and get qualitatively similar results. This should alleviate this concern.

fig 3. the labels 'heuristic' and 'conditional' seem not particularly discriminatory (TFT is arguably both heuristic and conditional, for example; I'm sure it has been described as both repeatedly in the literature) and the way they were applied was opposite to what I felt natural. Perhaps some other less ambiguous labels can be found.

We agree with the reviewer that the term 'conditional' was confusing in our previous manuscript. We had originally chosen it because it is relatively simple and intuitive, but upon reconsideration we agree that the risk of confusion is too high. We have opted to replace the term with the term 'context-dependent'. This is perhaps somewhat more technical-sounding, but will avoid confusion.

Reviewer #3

This is a paper on an interesting topic – the evolution of cooperative heuristics. I associate this idea with David Rand's group (e.g. Rand 2014), which they cite. The idea is that people play games with each other, but are uncertain about what games they are playing. They, therefore, develop heuristics that plays some average of the games. I think this has the potential to be an interesting model and publication.

However, I have some concerns about how the model is implemented and make various suggestions for improving it and conducting simpler version of the model to put their findings into context. I especially think that the authors need to spend some time explaining their key result. Heuristics evolve, but they do not explain *why* heuristics evolve. I have my suspicions, but the authors need to do a little more work to help the reader understand what is going on with their model.

I have some concerns about implementation of the model that may be giving misleading results. I am a modeler and understand that there are a vast number of ways to implement a model and decisions must be made. I get annoyed when reviewers suggest what I see as unnecessary complications or changes to one of my models. Therefore, when suggesting something different or more complicated, I will do my best to explain why I think it is necessary or important. The authors may find a different way to address my concerns that I did not consider.

I recommend a major revision to include additional simulations.

We wish to thank this reviewer in particular for the extensive review report and many valuable suggestions. We have made a number of major changes to the model based on this reviewer's suggestions (see below). Our results and conclusions remain qualitatively the same in spite of these changes.

Major comments:

1. Since it is the main finding of the paper, the authors really need to explain why heuristic strategies evolve in their simulations. By what mechanism do they out-compete conditional strategies? What is the selective advantage of using a heuristic strategy over a conditional strategy? Since this is the

whole point of the paper, the authors really need to explain it. Ideally the authors would do this as analytically or numerically as possible. For a given set of parameter combinations, what is the payoff to a conditional strategy and a heuristic strategy?

We agree with the reviewer that the previous version of the manuscript did not go into explanations of the results enough (we had originally prepared this manuscript for *Nature*, so we had very little room to elaborate). Having said that, because of the large genotype space (17 loci, four of which are continuous) it is not easy to get a comprehensive picture of exactly why evolution favours the strategies that it does in our model. However, there are some more approximate ways to obtain insight into why our simulations turn out the way they do. We have updated the manuscript with a two-fold approach to achieve this.

First, we have taken the reviewers' above suggestion, and analyzed the performance of the most commonly evolved heuristic and context-dependent strategies when played out against each other, for various levels of uncertainty. This gives us a rough idea of how likely both strategies are to emerge as dominant, depending on uncertainty (although it is far from a complete picture, because the strategy space is infinitely larger than the two strategies considered here). Interestingly, in most of the uncertainty range, neither strategy is vulnerable to invasion of the other strategy. However, it is clear that the invasion fitness of the heuristic strategy increases with uncertainty, whereas the invasion fitness of the conditional strategy decreases with uncertainty, which affects invasion chances in a finite population. These results are presented in a new section of the Supplementary Information (Section 3) and discussed in the Results section (lines 139–150) as well as (briefly) in the Discussion section (lines 185–188).

Second, as the reviewer suggests under point 7 (below), we have run benchmark simulations for a version of the model without any heterogeneity in social interactions, for a range of values of c . From this we obtain a number of insights – one of them is that evolution seems to favour strategies that are tuned to the 'average game' when evolution is run for this context in isolation (as the reviewer has suggested). We present these results in a new section in the Supplementary Information (Section 4) and discuss them in the Results section (lines 151–158) and the Discussion section (lines 180–183 and 189–200).

2. I suspect that the real reason that heuristic strategies evolve is because the authors implement an external cost to using a conditional strategy. This is somewhat buried in the appendix: "Individuals with a conditional strategy paid a fitness cost of 1%." This important fact needs to be moved into the main body of the paper. I think it should also be the authors' working hypothesis as to why heuristic strategies evolve unless they have a better explanation. This is not that exciting of an answer, though. Imposing an external cost on a strategy is an easy way to get it to evolve less often. The authors should be able to tell how much this cost matters either analytically or numerically. The authors need to move this information to the main body of the paper and discuss whether this is what is driving the evolution of heuristic strategies.

We agree that the cost of conditionality potentially obscured the results in the original model. In the new version of the model (which is the model now presented in the main text), there is no cost of conditionality at all anymore. Despite this, our conclusions remain the same.

3. The authors describe this external cost in the appendix. "This small cost for using a conditional strategy was implemented for practical reasons: if there would be no such penalty, there is the possibility that conditional strategies evolve to be de facto heuristic (if the threshold gets very close to -3 or 1), obscuring an appropriate analysis of the degree to which heuristics are used." I do not think this is the most parsimonious solution to this problem. Since this is a simulation model, the authors have access to the internal states of their agents. They should be able to tell if the threshold gets very close to -3 or 1 and just report it. Similarly, a de facto heuristic strategy can evolve if both of the conditional strategies are the same. It should be easy for the authors to check for this in their code. The authors, at a minimum, should run the simulations without the external cost to conditional strategies and see how it changes their results. I suspect that the heuristic strategy would have a hard time evolving without penalizing the alternative.

In the new model, in which there is no longer a cost of conditionality, we indeed classify 'de facto' heuristic strategies in the way that the reviewer suggests.

4. Another option would be for the authors to just be up-front about a cognitive cost driving heuristics, if that is what is driving their results. This might not be as interesting as getting their results without a cognitive cost, but it is still interesting.

This is now irrelevant, since we removed the cost of conditionality.

5. The authors need to explain this better "To avoid computational problems stemming from negative payoffs, a constant was added to each of the payoffs so that the lowest possible payoff was equal to zero." Does this mean that a 3 was added to all elements of the payoff matrix when $c = -3$ and nothing is added to the payoff matrix when $c = 1$? Or is 3 just added to all of the elements of all of the matrices? Either way, this needs to be more explicit than the vague statement in the appendix. The authors should just modify their discussion of the payoff matrices on Figure 1a with the actual values to include this bit that was added. I am not very comfortable with relative payoffs this close to zero. This implies that social interactions are all that matter to fitness. Normally, in evolutionary game theory, we implement a "baseline fitness" to take care of negative numbers and zeros. Baseline fitness is also useful because evolution maximizes the geometric mean fitness and thus really tries to avoid zero fitness. This can lead to all sorts of asymmetries with the specific values in the payoff matrix. Generally, the higher the baseline fitness the better the estimate of the direction of selection, but the slower selection occurs. In summary, to avoid negative numbers, the authors should just shift the payoff matrices to all positive numbers. I also think that they should implement a baseline fitness. They would need to try different values to see what gives them a good balance, but something like a baseline fitness of 50-100 would probably work (with expected payoffs from the games running from 0 to 60 - ~10 rounds with scores of 0 to 6 per round).

We thank the reviewer for this valuable suggestion, and recognize the fact that a very low baseline fitness can lead to some artefacts related to geometric mean fitness maximization. In the new version of the model, we have taken the reviewer's suggestion and implemented a baseline fitness of 100 (this is mentioned in line 226 in the Methods section). Our results remain qualitatively the same.

6. I am also uncomfortable with the way that the authors implement uncertainty. $cc_{pp} = cc * (1 - \mu\mu) + \varepsilon\varepsilon * \mu\mu$ This results in the player's observation of the cost of cooperating being uniformly distributed and not even centered on the observed value. I do not think there is any implementation of signal-detection theory that would use this distribution. This might really change the results by because it not only gives the player a noisy signal, but a biased and mostly uninformative signal (Figure 1). Figure 1. Current distribution from which the observed value of c (cp) is drawn when $c = 0$ (red dashed line) and $\mu = 0.5$. The distribution is uniform and not centered on c . This might makes the signal worse than it should be and therefore, heuristic strategies are likely less useful than they would be with a more realistic signal distribution. It is much more reasonable if the mode of cc_{pp} was c with likelihood decreasing away from cc_{pp} . The authors should change to this type of probability distribution to make sure that their results are not due to misleading observations. The authors could implement this by setting cc_{pp} as a random number drawn from a beta distribution where $\alpha\alpha$ is the uncertainty parameter (from 1 to ∞) and $\beta\beta = 4(\alpha\alpha - 1) cc + 3 + 2 - \alpha\alpha$ that is then multiplied by 4 and added to -3. All of these 4s and 3s are in there to keep cc_{pp} on the interval from -3 to 1 (see Figures 2-4). Figures 2-4. An implementation using the beta distribution as described in this review. Now the modes are at the actual value of c (red dashed line) and the likelihood of the signal falls as one gets away from c . These are for $\alpha = 10, 100, 1000$. $\alpha = 1$ gives a uniform distribution over the whole range (similar to $u = 0$ in the submitted version). I've appended the code for generating these figures at the end of this review. This would aid the authors in implementing a more reasonable error function.

Again, we thank the reviewer for this valuable suggestion. We have implemented the beta distribution with mode-centering around the actual value of c (as the reviewer suggests). In addition, we have made sure that the variance of the beta distribution remains constant along the entire range of c , for

the same level of uncertainty. This is necessary, because if one simply keeps alpha constant, and moves beta to obtain a distribution with the required mode, the variance actually changes, leading to a situation with different uncertainty. Hence, both alpha and beta have to be updated along the range of c to obtain the distribution with the required mode and variance. In the new model, uncertainty is (linearly) proportional to the variance of the beta distribution, with uncertainty=0 corresponding to a situation with 0 variance (in practice, extremely small variance), and uncertainty=1 corresponding to a situation where the beta distribution is in fact a uniform distribution over the entire range (so completely uninformative). Despite this new implementation, our results remain qualitatively the same.

7. The authors need to show how their results differ from null models. This can be done in the appendix and referenced in the main paper. The first null model would be one where there is no variation in the payoff matrix. The reader should be able to compare the results of the more complicated models to the simpler models. They should run their model with constant game matrices at various values of c . Half unit intervals (i.e., $c = -3, -2.5, -2, -1.5, -1, -0.5, 0, 0.5, 1$) would be sufficient. What strategies evolve under each of the the steady-state conditions? How much cooperation is there? This will let the reader see to what extent the results in, say, Figure 2b and 3 are an average of the steady-state games. It also will give a robustness check. There would not need to be any heuristic strategies in this model since there is no uncertainty over the game type.

As explained under point (1), we have taken this suggestion and added a section to the Supplementary Information (Section 4) where we show these results (with even higher resolution than the reviewer suggests). It is indeed the case that the results we get for maximal uncertainty are very close to (yet not identical to) the evolution of strategies in the average game. We discuss these results in the Results section (lines 151–158) and in the Discussion section (lines 180–183).

8. Another simpler model that the authors should report (in the appendix, referenced in the paper) would be one where players only use strategies of pure cooperation or pure defection. This would, again, give readers the ability to see whether the strategies are just averaging over the games. The authors would implement both conditional and heuristic strategies in this version of the paper. This would give the readers a baseline with which to compare the more complicated strategies. I actually think the paper would be more convincing if they just used these simple strategies instead of introducing the more complicated repeated game structure. It would be a more tractable implementation of their idea. Another advantage of this simpler version of their model is that the authors should be able to give an analytic analysis of the expected payoffs for cooperating or defecting under various values of c_p with given values of μ . They could even plot these and the reader could compare these to their results from the more complicated model.

By simplifying the strategy space to simple cooperation or defection without the possibility to condition on earlier game rounds, the game, even if it is still repeated, is in essence reduced to a one-shot game. Individuals can now only cooperate in all rounds or defect in all rounds, which is essentially the same as playing a single game with higher payoffs. As we know, defecting is the dominant strategy in a one-shot Prisoner's Dilemma game, so we should expect the evolution of strategies that always defect if $c < 0$, and always cooperate if $c > 0$, as long as uncertainty is low enough (leading to a cooperation rate of around 0.25). For high uncertainty we should expect a heuristic that only defects, because the average game is a Prisoner's Dilemma game. We ran the simulations and this is exactly what we find, although context-dependent strategies tend to become somewhat more conservative with increasing uncertainty (they only switch to cooperation with somewhat higher c). Although we could in principle have reported on these results in the SI, we have decided against this because the results are somewhat trivial, and adding this would lead to a cluttering of the information presented in the SI. However, we provide some graphs below so that the reviewer can inspect the results (they are 100 replicates per uncertainty level). In the graph that shows the evolution of the gene that determines the point at which the individuals switch from defection to cooperation (T ; rightmost panel), the transparency of the points is linked to the frequency of context-dependent strategies in the

simulation. This is because the switchpoint is not relevant (i.e. not phenotypically expressed) for heuristic strategies. The code is available on github (see below).

9. If the authors want to have a more complicated strategy space, I am uncomfortable with them using strategies that only remember the previous interaction. The reason is that the invasion dynamics for repeated games are complicated, these complications only arise with forgiving strategies of memory two or greater, and focusing only on memory one strategies can be misleading. This is a bigger deal because the authors' conclusion is that their "model provides insight into why individuals often behave more cooperatively than standard evolutionary and economic theory would predict." The reasons for these complications are the folk theorem of repeated games and findings that there is no evolutionarily stable strategy in the iterated prisoner's dilemma. For example:

Boyd, Robert, and Jeffrey P. Lorberbaum. "No pure strategy is evolutionarily stable in the repeated Prisoner's Dilemma game." *Nature* 327.6117 (1987): 58-59.

As an example of where overly simple strategies can lead a paper wrong, the authors should look at a paper they cite:

Delton, Andrew W., et al. "Evolution of direct reciprocity under uncertainty can explain human generosity in one-shot encounters." *Proceedings of the National Academy of Sciences* 108.32 (2011): 13335-13340. A critique pointed out that adding slightly more complicated strategies could destroy cooperation in the exact same model:

Zefferman, Matthew R. "Direct reciprocity under uncertainty does not explain one-shot cooperation, but demonstrates the benefits of a norm psychology." *Evolution and Human Behavior* 35.5 (2014): 358-367.

Plus additional commentary of interest:

Delton, Andrew W., and Max M. Krasnow. "An independent replication that the evolution of direct reciprocity under uncertainty explains one-shot cooperation: Commentary on Zefferman." *Evolution and Human Behavior* 35.6 (2014): 547-48.

Zefferman, Matthew R. "When repentance and forgiveness are possible, direct reciprocity does not explain one-shot generosity, even under arbitrarily high levels of uncertainty." *Evolution and Human Behavior* 35.6 (2014): 548-49.

I worry that the authors are introducing a similar problem in their model as the Delton and Krasnow paper. To make a stronger claim about the implications of their model to human cooperation, the authors should implement their model with strategies of at least memory two. Note: I know that it may seem contradictory to both use simpler and more complicated strategies. Specifically, I am suggesting that simpler strategies would make their findings about the evolution of heuristics more tractable.

More complicated strategies are needed to make their findings about the evolution of cooperation more believable.

We appreciate this concern. However, we would like to argue that the issue that arises in the paper of Delton, Kranow, Cosmides & Tooby (DKCT) is very different from anything that could arise in our model. In DKTC, individuals have uncertainty about whether they are playing a one-shot game or a repeated game. Only two strategies are allowed: TIT FOR TAT (always cooperate in the first round, then copy opponent's move) or ALLD (always defect). The authors find that if there is high uncertainty about whether the game individuals are playing is one-shot or repeated, TIT FOR TAT evolves. Since TIT FOR TAT by definition cooperates in the first round, this leads to cooperation if the game is one-shot. We completely agree with Zefferman's criticism of this setup: the DKCT results can be explained by their artificial restriction of the strategy space – the only allowed cooperative strategy always cooperates in the first round, so it has no choice but cooperating in one-shot interactions. As Zefferman then nicely shows, allowing strategies that defect in the first round, but repent later, completely reverses the results (because these strategies then evolve under high uncertainty, and defect in one-shot games).

Our set-up is very different. First of all, we are not interested in uncertainty about whether a game is repeated or one-shot, but about the payoffs of the (repeated) game. So the very specific artefact that occurs in DKCT (only allowing a cooperative strategy that cooperates in the first round, leading to one-shot cooperation), cannot occur in our simulations. Second, our strategy space is much less constrained than in DKTC. DKTC only allow two handpicked strategies (TFT and ALLD), whereas our strategy space is confined by principles (one step of memory, continuous first move loci, continuous threshold locus, and a switch between context-dependent and heuristic strategies). As a result, our strategy space is much larger. The size of our strategy space was a conscious design choice: we didn't want to arbitrarily confine it too much (as in DKTC), but we also wanted it to have a size that evolution can still feasibly search. Our strategy space allows for strategies that defect on the first move and repent later. For example, the strategy DIMAS from Zefferman's paper is in fact part of our strategy space and could potentially evolve.

Apart from the fact that we don't believe the problem in DKTC applies to our simulations, we also don't think expanding the strategy space to allowing strategies with two steps of memory would be a prudent step. The reason is that this would completely explode the size of the strategy space, which would not only make evolution very inefficient in searching it (so outcomes would become highly path-dependent, and general principles would be very hard to infer), but also make the model very hard to compare with our original model. To give an idea, our original model contains 17 loci, of which 13 are Boolean (3 substrategies of 4 loci, and 1 'master switch' between the heuristic and context-dependent strategy). This means there are $2^{13} = 8,192$ possible genotypes for the Boolean loci alone. Allowing two steps of memory would increase the size of each substrategy to 16 Boolean loci (because there are $4 * 4 = 16$ possible histories when looking two steps back). This means we would need a total of $16 * 3 + 1 = 49$ Boolean loci, leading to $2^{49} = 5.6 * 10^{14}$ possible genotypes. This is a difference in size of 11 orders of magnitude. For these reasons, we have decided against running simulations with 2 steps of memory.

Of course, defining the strategy space always comes with choices. Any choice will make it impossible for some strategies to evolve, that might have evolved if they had been allowed. We do not claim that our choices of confining the strategy space do not influence the outcome of evolution in our model, and may in some ways bias the results. But this is true for any evolutionary model. For us, the most important consideration has been to allow significant freedom (not handpicking allowed strategies), but to keep the strategy space confined enough that evolution can feasibly search it and therefore lead to reasonably consistent results. Having said all this, we fully share the basic concern of the reviewer that these choices are very important, and can be crucial in shaping the results. We have therefore dedicated an entire paragraph of the Discussion section to discussing these design choices, including the size of the strategy space (lines 201–218).

10. The authors should provide their code and results in a publically accessible data and code archive, such as GitHub, BitBucket or the Open Science Framework. The authors should report, in the paper, where their code and data is or will be stored.

We have made our code available on github (https://github.com/pvdberg1/uncertainty_cooperation_evolution) and our data is available upon request (we have not uploaded the data in a repository because it is easily reproducible using the code and very large [about 50 GB], but would be happy to share these data with the reviewer personally already now). We have added code and data availability statements to this effect in the manuscript (lines 252–256).

More minor comments:

11. The set-up for their model is very similar to “multiple games theory” papers by Bednar and co-authors. In these games, individuals play iterated games with uncertain payoff matrices. They talk about this in terms of heuristics and there is even an experimental literature on this. The authors could start with the below papers to familiarize themselves with this literature. Should acknowledge and address similarities in the models and ideas in these papers and their submission.

Bednar, Jenna, and Scott Page. "Can game (s) theory explain culture? The emergence of cultural behavior within multiple games." *Rationality and Society* 19.1 (2007): 65-97.

Bednar, Jenna, et al. "Behavioral spillovers and cognitive load in multiple games: An experimental study." *Games and Economic Behavior* 74.1 (2012): 12-31.

Grimm, Veronika, and Friederike Mengel. "An experiment on learning in a multiple games environment." *Journal of Economic Theory* 147.6 (2012): 2220-2259.

We thank the reviewer for directing us to this literature. The papers are relevant – we now cite them (lines 32–34) – but the setup in these papers is quite different from ours. Although the Bednar papers do deal with heterogeneity in social contexts, there is no uncertainty about these contexts with the agents. The model in the first paper focuses on constraining the cognitive architecture – basically only allowing very limited flexibility in strategies for different games. In this sense, the heuristic nature of the strategies in this paper are a direct result of the assumptions on the cognitive machinery, rather than an outcome of the model. This is also not the focus of our paper; our main focus is how these constraints can lead to different behaviours when individuals operate in different worlds (where different interaction contexts can come up). The second paper is an experimental paper based on this framework. The third paper is an experimental paper that does apply different ‘information levels’ about the game at hand, but this is not done by introducing uncertainty about the payoff, but rather by varying how long participants get to see the payoff matrix for. So although we agree that these papers provide some important background and introduction to our model, we do not think that they merit an extensive discussion in our manuscript.

12. The authors should justify randomizing the number of interactions. Why not make things simpler by keeping them constant at 10? Since they are only using strategies with memory one (or two), there is no danger of backwards induction.

This is a good point. We have changed this in the current model (this change does not qualitatively affect the results).

Figure generating R code:

```
# Uniform distribution used in paper
```

```
c = 0
```

```
u = .5
```

```

cp_distribution = c * (1-u) + runif(10000000, min = -3, max = 1)*u
hist(cp_distribution, breaks=100, xlim=c(-3,1))
abline(v=c, lty=2, col="red", lwd=3)

# Suggested beta distribution

c= 0
mode = (c + 3)/4
alpha = 1
beta = (alpha - 1)/mode + 2 - alpha
cp_distribution = 4*rbeta(10000000, shape1=alpha, shape2=beta, ncp = 0)-3

hist(cp_distribution, breaks=100, xlim=c(-3,1))

abline(v=c, lty=2, col="red", lwd=3)

```

Reviewer #4

In this paper the authors study, using a simulation model, the evolution of cooperative behaviour when there are uncertainties in the types of social interactions that can arise. The payoff structures vary in the different types of interactions considered, ranging from situations where cooperation is always the best action for the focal individual (regardless of the behaviour of the interaction partner), to versions of the Prisoner's Dilemma, to situations where it never pays to cooperate. The authors attempt to study how the degree of uncertainty that individuals have about the nature of the social interactions affects: (1) the evolution of heuristic decision making, and (2) the level of cooperation in the population. They find in their simulations that uncertainty can lead to the evolution of simple heuristic strategies that disregard the available information about the nature of the social interaction, and In addition, that these heuristic strategies are more cooperative than the strategies that evolve when individuals are better able to assess social interactions.

Studying the effects of uncertainties in the types of social interactions on the evolution of cooperation is an potentially interesting topic, however, the result found in the current paper are rather trivial. It is essentially obvious, due to the way the authors have defined their model, that increased uncertainty will result in increased use of heuristic strategies.

We are not entirely sure what aspect of the way we defined our model the reviewer is referring to, but it is possible that (s)he is referring to the fact that we implemented a cost of conditionality that context-dependent strategies had to pay. This indeed aided the evolution of heuristic strategies in the previous version of the model. This is a good point, and we have therefore entirely eliminated this cost of conditionality from the current model. Despite this, the results remain qualitatively the same.

Another possibility is that the reviewer refers more to the broad set-up of the model. If so, we agree with the reviewer that it is intuitively to be expected that heuristic strategies will perform better with increased uncertainty. Since the reliability of information decreases with increasing uncertainty, strategies that base their behaviour on this information (the context-dependent strategies) will fare worse. However, although it might be expected that heuristic strategies will outperform context-dependent strategies under full uncertainty (when information is essentially worthless), it is not immediately obvious that they will do so for intermediate uncertainty, and for what level of uncertainty we will start seeing them emerging (i.e., we could not have predicted the shape of Figure 2b from only knowing the set-up of the model). Therefore, we think that our main message is still valuable: even intermediate levels of uncertainty lead to the evolution of social heuristics, this significantly impacts cooperation levels, and leads to behaviours that seem suboptimal when considered in isolation.

Moreover, it is clear that the level of cooperation may increase, since the authors are allowing for situations in which cooperation is always the best action.

We do not fully understand why the reviewer considers it obvious that high cooperation levels should evolve under higher levels of uncertainty. The reviewer is correct that there is a region of the parameter space where cooperation is always the best action, but there is also an (equally large) region where defection is always the best action. In addition, the level of uncertainty does not affect the probability with which the individuals in our model encounter an interaction in which cooperation is always the best action – this probability is 25%, regardless of the level of uncertainty. So it is not clear to us why the reviewer expects cooperation to increase with increasing uncertainty, and we do not think that this was obvious *a priori*. Having said that, the reviewer (together with some of the other reviewers) has convinced us that our previous manuscript was lacking discussion about the reasons why we get the outcomes that we get. This is why we have now included two full sections with additional simulations to the Supplementary Information to provide additional insight into our main results (Sections 3 and 4; also see next point of this reviewer). We discuss these in the Results section (lines 139–158) and the Discussion section (lines 180–200).

The more interesting question would be how the frequency of cooperation depends on the various types of payoff structures that are encountered, and the authors have not analyzed this important question in any detail.

This is an important point. We have taken the reviewer's suggestion, and have now added an entire section to the Supplementary Information in which we systematically check the outcome of evolution for various payoff structures (Section 3). We use these results as a benchmark with which the results of our main model can be compared to, and refer to them in the Results section (lines 139–150) and (briefly) in the Discussion section (lines 185–188).

An additional, and potentially serious, shortcoming of the approach adopted here is that only deterministic strategies are considered. This assumption is at odds with the authors' notion that the individuals are interacting in an uncertain world — in such a world errors due to the mis-implementation or mis-reading of strategies would be expected to be common, and thus, the more natural framework to use would be that of stochastic strategies. Some of the key results that the authors find in their simulations may be quite different with stochastic strategies. For example, GRIM was found to do well in the simulations (since it cooperates with itself), however, this would almost certainly change if the strategies allowed for errors, since an occasional mistake would then result in long stretches of mutual defection.

We now realize that the previous structure of our manuscript, in which most methodological details were relegated to the Supplementary Information, has caused some confusion about the set-up of our model. The fact is that our original model already did include errors (as does the current version of the model): individuals mistakenly implement the wrong behaviour with a probability of 1% (*i.e.*, they cooperate if their strategy dictates defection, or *vice versa*). The reason that GRIM still does relatively well in our simulations is that the length of a single repeated interaction is relatively short in our simulations (10 interaction rounds – we do not attempt to approach infinitely repeated interactions), so an accidental defection in a grim-grim interaction does not have a very high cost in terms of foregone rounds of mutual cooperation. In the current version of the manuscript, we explicitly explain this in the Results section of main text, where we introduce the model (lines 102–103). We now also mention that GRIM can evolve despite these errors because the repeated interaction length is relatively short (lines 130–133).

Finally, the format of the paper is unsuitable for Nature Communications — the body of the paper is very brief, with most of the important details relegated to the Supplementary Information.

We fully agree with the reviewer. The reason was that the paper was originally written up for *Nature*, and then later transferred to *Nature Communications*. We have now significantly expanded the manuscript into a format that is appropriate for *Nature Communications*. We have moved all of the methodological details to the main text. We now explain the model in much more detail at the start of the Results section (*Nature Communications* does not allow a separate 'The Model' section between the Introduction and the Results), and moved additional details of the model to the Methods section at the end of the manuscript. We have also significantly expanded the Discussion section. The Supplementary Information now mostly contains additional analyses.

REVIEWERS' COMMENTS:

Reviewer #2 (Remarks to the Author):

I am happy that the authors have responded appropriately and constructively to the points raised by myself and the other reviewers.

Reviewer #3 (Remarks to the Author):

At the start, I should say that I am sorry that I do not have the capability at this time to read the revised manuscript in as much detail as the original and am also late with my second review.

However, I have read the authors' response to the reviewers' comments with much interest and am satisfied that they have addressed enough of my concerns and made sufficient revisions to the model for publication.

I hope that the authors' feel that their model and manuscript has been improved by the peer review process. I know it can be annoying to do additional analysis at a reviewer's request, but I think their results are stronger for it.

Reviewer #4 (Remarks to the Author):

The authors have made a serious effort to revise the manuscript and the resulting paper is significantly improved. My previous concerns have been addressed and I consider the paper to now be suitable for publication.